# Predictive factors for limited health literacy among persons with cirrhosis: A Swedish explorative cross-sectional study

**Maria Hjorth**[1,2¤a]*, **Anna Forsberg**[3,4¤b]

**1** Centre for Clinical Research in Dalarna, Falun, Sweden, **2** Department of Public Health and Caring Sciences, Uppsala University, Uppsala, Sweden, **3** Institute of Health Sciences at Lund University, Lund, Sweden, **4** Department of Cardiothoracic Surgery, Skåne University Hospital, Lund, Sweden

¤a Current Address: Centre for Clinical Research in Dalarna, Falun, Sweden
¤b Current Address: Institute of Health Sciences at Lund University, Lund, Sweden
* maria.hjorth@regiondalarna.se

## Abstract

### Introduction

Fatigue and altered cognitive capacity are common symptoms following cirrhosis. Patients consider information about cirrhosis difficult to understand. Health literacy levels vary among persons with chronic illnesses, which can hamper participation in and adaptation to treatment, potential restrictions and recommendations. Limited health literacy might also lead to decreased autonomy.

### Purpose

The aim was to explore predictors of limited health literacy among adults with cirrhosis.

### Materials and methods

This cross-sectional study explored health literacy among 167 Swedish adults with cirrhosis, 94 men and 73 women with a median age of 65 years using the 'Newest Vital Sign' instrument. Predictors of limited health literacy were examined in relation to patient characteristics and cirrhosis disease events. The study is reported following the STROBE guidelines.

### Results

The prevalence of limited health literacy was 58%. Low education and covert hepatic encephalopathy were associated with limited health literacy ($p < 0.05$).

**Data availability statement:** The data contains sensitive patient information and cannot be shared publicly because of ethical restrictions decided by the Swedish Ethical Review Authority and the data register holders. Following approval by the Swedish Ethical Review Authority, the de-identified dataset analysed in this study can be made available from Region Dalarna, upon request to e-mail forsknings.utlamnande@regiondalarna.se

**Funding:** This work was funded by Uppsala University, Ester Åsberg Lindbergs Foundation and the Centre for Clinical Research in Dalarna. The equipment for continuous reaction time and enforcement of the intervention nurses' tutorial group sessions was funded by Norgine. The sponsors had no impact on the study design, data collection, analysis or interpretation of data, reporting data or submission for publication. Open access funding provided by Uppsala University.

**Competing interests:** The authors have declared that no competing interests exist.

## Conclusion

Limited health literacy is common among Swedish adults with cirrhosis. Both covert hepatic encephalopathy and low education might be predictors of limited health literacy. Healthcare providers should tailor their patient education based on the patient's literacy level to facilitate understanding, learning and self-management.

## Introduction

Cirrhosis is a collective term for the end-stage of scarred liver tissue, which results from a chronic liver disease [1]. Early symptoms are generally vague, e.g. fatigue [2]. Minimal cognitive changes may occur, sometimes defined as covert hepatic encephalopathy [3–5]. Covert hepatic encephalopathy may predict serious events with overt hepatic encephalopathy [6]. The advanced cirrhosis symptoms, i.e. ascites, overt hepatic encephalopathy or gastro-oesophageal variceal bleeding, are driven by portal hypertension and systemic inflammation. These serious symptoms limit patients' survival and impair their quality of life [1]. Symptom distress increases the need for medical treatment and self-management support [7,8].

Patients with cirrhosis are expected to participate with healthcare professionals (HCPs) to optimise their health [8]. The patients' individual level of health literacy has an inevitable impact on their understanding of the disease and ability to participate [9]. Health literacy is defined as personal skills that enable a person to obtain, understand, assess and use information to make decisions and act to promote health [10]. Patients with limited health literacy levels have increased healthcare costs, which are associated with: a greater number of hospitalisations [11]; difficulties communicating with healthcare providers; low knowledge of the disease [9]; poorer adherence to medical recommendations [9,11]; and reduced ability to interpret health information texts [11]. In comparison with persons with other gastrointestinal diseases, e.g. inflammatory bowel disease and gastrointestinal cancer, limited health literacy tends to be more common following cirrhosis [12]. In other European countries, limited health literacy among persons with cirrhosis have been associated with male gender, low education [13], impaired liver function, covert hepatic encephalopathy, ascites, falls and depressive symptoms [14].

Patients experience cirrhosis disease information presented by HCPs or on the internet as complex [13,15]. They also express that collaboration with HCPs is vital. However, they do not always feel involved [16]. Poor understanding of the cirrhosis and lack of self-management skills lead to repeated inpatient care, some of which may be preventable [17,18]. Patients wish to have information presented in a way that facilitates their learning and understanding of the disease [16,19]. Hence HCPs need to be aware of individuals who may be at risk of not acting upon the advice received. Although health literacy has been explored among persons with cirrhosis in other European countries [13,14], it may vary among different nationalities [10]. Therefore, as a baseline measure in a multicentre, randomised controlled study of nurse-led clinics for patients with cirrhosis, health literacy data were collected from

a Swedish cohort of adults with cirrhosis [20]. The aim of the present study was to explore predictors of limited health literacy among adults with cirrhosis.

## Materials and methods

### Design

This explorative, cross-sectional study employed a quantitative approach to search for predictors of limited health literacy among Swedish adults with cirrhosis. The STrengthening the Reporting of OBservational studies in Epidemiology (STROBE) checklist provided guidance in reporting the study [21] (S1 Table).

### Participants and setting

Six hepatology clinics were involved in patient recruitment in the multicentre, randomised controlled trial (RCT) [20].The clinics, two county hospitals and four university hospitals, were situated in mid and south Sweden. All participants had received regular consultations by hepatologist since the time for cirrhosis diagnosis. Diagnosis of cirrhosis was based on clinical signs, laboratory findings, histology, magnetic resonance imaging, computer tomography, ultrasound and/or transient elastography. Overt hepatic encephalopathy was examined according to the West Haven criteria [3]. Inclusion and exclusion criteria were examined by a hepatologist before recruitment (Table 1). One or two registered nurses per study site were trained and thereafter responsible for patient recruitment, informed consent and data collection.

### Data collection

Data were collected at each hepatology clinic from 17th of November 2016–31st of December 2020 as patients entered the above mentioned RCT [20]. The total time for data collection was 30–45 minutes per patient.

**Assessing health literacy.** Functional health literacy was measured with the Newest Vital Sign (NVS) generic instrument [22], which comprises six standardised questions about information on a nutrition label. When the patient had studied the nutrition label, the registered nurse asked six questions about its content. The nutrition label was available to the patient and could be referred to during the procedure. The registered nurse recorded each response on a score sheet that contained the correct answer. A correct answer scored one point, resulting in a total score from zero to six. Four correct answers is the cut-off between a possible limited health literacy (score 1–3) and a normal health literacy (score 4–6). Cronbach α for the English version is .76 [22]. The instrument had previously been translated from English to Swedish [23].

**Predictive factors.** The predictive factors examined in this study were: patients' background characteristics and cirrhosis related disease events. During the interview patients were asked about their comorbidities and background characteristics, such as age, gender, level of education and work ability. Information about cirrhosis diagnosis, overt

**Table 1. Inclusion and exclusion criteria for study participation.**

| Inclusion criteria | Exclusion criteria |
| --- | --- |
| Age 18–85 | Persistent overt hepatic encephalopathy |
| Cirrhosis diagnosis within 24 months | Inability to read or communicate in Swedish |
| Ongoing follow-up by hepatologist | Severe comorbidity: Chronic obstructive pulmonary disease grade 3–4 Coronary heart disease New York Heart Association Functional Classification (NYHA) classes 3–4 Dementia Actual advanced cancer Stroke with sequelae Severe psychiatric disease Renal failure requiring dialysis |

hepatic encephalopathy events, ascites and oesophageal variceal bleeding were collected from medical records. Blood sampling included analyses of the MELD-score [24] and Child Pugh score [25].

**Identifying covert hepatic encephalopathy.** The presence of covert hepatic encephalopathy was detected by use of the psychometric encephalopathy score (PHES) [26] and continuous reaction time (CRT) [27]. PHES consists of a five-step paper and pencil test including two number connection tests, a digit symbol test, a serial dotting test and a line tracing test. PHES examines the patient's motor speed and accuracy, visual perception, visuospatial orientation, visual construction, concentration and attention. The total PHES score ranges from +6 to −18. A score of -4 or lower is the cut-off for a pathological result [26]. CRT is a test with auditory stimuli in headphones, which emits 150 signals in intervals from one to six seconds. CRT examines the reaction time and endurance by pushing a trigger button after a signal. Using the EKHO software reaction-time analysis tool, an index <1.9 separates covert hepatic encephalopathy from other brain dysfunctions with a specificity of .92 and sensitivity of .93 [27]. According to previous recommendations [3], abnormal test results from both PHES and CRT are required for diagnosing covert hepatic encephalopathy.

## Statistical analysis

Continuous data are reported in medians with interquartile ranges (IQR). Categorical data are reported in frequencies and percentages. A generalized logistic regression model was used for two explorative analyses of predictive factors for limited health literacy, defined by a NVS score ≤3. Risk ratio (RR) and confidence intervals (CI) of 95% were reported for each factor. The first model included factors pertaining to the following patient characteristics: age (≤64 *vs* >65); gender (female *vs* male); level of education (upper secondary school or higher *vs* elementary school); other comorbidity (no *vs* yes); and reason for cirrhosis diagnosis (other cirrhosis diagnosis *vs* alcohol related cirrhosis). The second model included cirrhosis disease event factors, i.e. ability to work (yes *vs* no); MELD-score (≤10 *vs* ≥11); presence of hepatic covert encephalopathy (no *vs* yes); previous event of hepatic overt encephalopathy (no *vs* yes); ascites (no *vs* yes), and oesophageal variceal bleeding event (no *vs* yes).

Jamovi (Version 2.6.11) [Computer Software] retrieved from https://www.jamovi.org was used for analysing data. Due to the exploratory design, no adjustments for multiple testing were made, $p < 0.05$ was considered significant. Only patients with complete datasets were included in the respective analyses.

## Ethics

The study was approved by the Regional Ethics Board in Uppsala, Sweden (Dnr 2016/146) and performed in line with the principles of the declaration of Helsinki [28]. Study participation was preceded by written informed consent.

## Results

In total, 167 adults, 94 men and 73 women with a median age of 65 years were included in this study, of whom 58% were identified as having limited health literacy, i.e. ≤3 points on the NVS. The most common cirrhosis aetiology was alcohol related liver disease. At the data collection time-point, 99 (59%) of the 167 study participants presented a stable cirrhosis disease stage, i.e. Child-Pugh group A, despite the fact that 109 participants (65%) had previously experienced symptoms of advanced cirrhosis disease, such as ascites, overt hepatic encephalopathy and/or bleeding from oesophageal varices. Of these symptoms, ascites was the most common, present in almost half of the participants (n = 79). Twenty-four had experienced a combination of ascites, overt hepatic encephalopathy and/or bleeding from oesophageal varices. The majority had no other comorbidities (n = 97). Additional background characteristics are presented in Table 2.

In the first generalized logistic regression model, predictive factors for limited health literacy were explored in relation to five background characteristics, based on data from 165 of the 167 participants. Low education, i.e. elementary school

**Table 2. Baseline characteristics of the entire study group.**

| Age, median (25–75 percentile) | 65 (57–70) |
|---|---|
| Male gender, n (%) | 94 (56) |
| Upper secondary school or higher education, n (%) | 127 (75) |
| Cohabiting, n (%) | 107 (65) |
| Born in Sweden, n (%) | 154 (92) |
| Occupation | |
| Student or working, n (%) | 48 (29) |
| Retired, n (%) | 82 (49) |
| Fulltime sick leave, n (%) | 11 (7) |
| Unemployed, n (%) | 26 (15) |
| Aetiology of liver disease | |
| Alcohol-related liver disease, n (%) | 86 (52) |
| Metabolic dysfunction associated steatotic liver disease, n (%) | 22 (13) |
| Autoimmune liver disease, n (%) | 18 (11) |
| Cryptogenic, n (%) | 29 (17) |
| Chronic viral hepatitis, n (%) | 7 (4) |
| Other, n (%) | 5 (3) |
| Characteristics of cirrhosis | |
| Child-Pugh A/B/C, n (%) | 99 (59)/57 (34)/11 (7) |
| MELD-score, median (25–75 percentile) | 9(8–12.5) |
| History of overt hepatic encephalopathy, n (%) | 21 (13) |
| History of ascites, n(%) | 79 (47) |
| History of oesophageal variceal bleeding, n (%) | 36 (22) |
| Current comorbidity | |
| Musculoskeletal, n (%) | 6 (4) |
| Gastrointestinal, n (%) | 7 (4) |
| Diabetes, n (%) | 27 (16) |
| Cardiovascular, n (%) | 16 (10) |
| Pulmonary, n (%) | 7 (4) |
| Other, n (%) | 20 (12) |
| Numbers of comorbidities | |
| None, n (%) | 94 (58) |
| One or two, n (%) | 63 (39) |
| Two or more, n (%) | 4 figure(3) |

level, was associated with limited health literacy (RR 1.49, 95% CI 1.0–2.2). In contrast, having another comorbidity correlated to a normal health literacy level, i.e. NVS ≥ 4 (RR 0.64, 95% CI 0.4–0.9) (Table 3). S2 Table 1, provides a complete and detailed description of the analysis.

In the second generalized logistic regression model, predictive factors for limited health literacy were explored in terms of cirrhosis-related disease events, based on data from 156 of the 167 participants. Limited health literacy was associated with covert hepatic encephalopathy, confirmed by psychometric tests (RR 1.54, 95% CI 1.0–2.3). A history of cirrhosis decompensation, i.e. ascites, overt hepatic encephalopathy or gastroesophageal bleeding, was not associated with limited health literacy in this study group (Table 4). S2 Table 2, provides a complete and detailed description of the analysis.

**Table 3. Logistic regression analyses presenting risk ratio of associations between patient characteristics and limited health literacy.**

| Predictor | Risk ratio | 95% Confidence interval (CI) | |
|---|---|---|---|
| | | Lower | Upper |
| **Age** | 1.20 | 0.81 | 1.77 |
| 18-64 *vs* 65–85 | | | |
| **Gender** | 1.14 | 0.76 | 1.70 |
| Female *vs* male | | | |
| **Education** | 1.49 | 1.02 | 2.19 |
| Upper secondary school/university *vs* elementary school | | | |
| **Comorbidity** | 0.64 | 0.41 | 0.99 |
| No *vs* yes | | | |
| **Alcohol related liver disease** | 0.84 | 0.55 | 1.27 |
| No *vs* yes | | | |

In the model, age ≤ 64; female gender; education in upper secondary school or higher; no other comorbidity; diagnoses other than alcohol related cirrhosis were assumed protective factors for limited health literacy.

**Table 4. Logistic regression analyses presenting the risk ratio of associations between cirrhosis related disease events and limited health literacy.**

| Predictor | Risk ratio | Confidence interval (CI) | |
|---|---|---|---|
| | | Lower | Upper |
| **Work ability** | 1.60 | 0.90 | 2.84 |
| No *vs* yes | | | |
| **MELD-score** | 1.19 | 0.81 | 1.76 |
| ≤10 vs ≥ 11 | | | |
| **Covert hepatic encephalopathy** | 1.54 | 1.03 | 2.31 |
| No *vs* yes | | | |
| **Overt hepatic encephalopathy** | 1.20 | 0.78 | 1.85 |
| No *vs* yes | | | |
| **Ascites** | 1.11 | 0.75 | 1.66 |
| No *vs* yes | | | |
| **Oesophageal variceal bleeding** | 1.05 | 0.67 | 1.65 |
| No *vs* yes | | | |

In the model, MELD-score ≤10; being able to work; no presence of covert hepatic encephalopathy; no episode of overt hepatic encephalopathy; no ascites; no event of oesophageal variceal bleeding were assumed protective factors for limited health literacy.

## Discussion

This is the first study of health literacy in a Swedish cirrhosis population. Health literacy is an insufficiently explored area after cirrhosis, which justifies the explorative design of the present study. The most important finding was the 54% higher risk of limited health literacy in patients with covert hepatic encephalopathy. Covert hepatic encephalopathy is evident in approximately 20–80% of patients with cirrhosis [3,4]. However, it is hardly visible without psychometric testing [3], and therefore commonly under-reported. Although it is known that covert hepatic encephalopathy may precede overt hepatic encephalopathy [6], it is not routinely screened for [5]. This result strengthens previous assumptions made by Kaps et al.

[14] and highlight patients' need of symptom relief of covert hepatic encephalopathy, which may improve self-management [9,11] and autonomy [11]. Hence, our recommendation is to implement regular screening for covert hepatic encephalopathy and adopt a strategy to reduce the symptoms.

Interestingly, we found no association between previous overt hepatic encephalopathy events and limited health literacy. Although there is a fourfold risk of persistent covert hepatic encephalopathy following an episode of overt hepatic encephalopathy [4], none of our patients had a residual risk of limited health literacy after this event. Accordingly, the present findings suggest that the level of health literacy may fluctuate over time in cirrhosis and depend on the patient's current symptoms.

The present study supports the finding of Freundlich Grydgaard and Bager [13] by confirming that low education is a predictor of limited health literacy in cirrhosis populations. This result is also in line with reports of limited health literacy in general populations [10]. In contrast with Kaps et al., [14], a higher MELD-score or symptoms of ascites did not predict limited health literacy. These contradictory findings may be explained by different MELD score classifications of patients into high or low MELD-score in the two studies. Nor could gender predict limited health literacy, which contradicts findings by Freundlich Grydgaard and Bager [13]. In our study, comorbidity had a positive influence on health literacy, which contradicts previous results of Freundlich Grydgaard and Bager [13], who found that comorbidity had a non-significant influence on health literacy. One reason might be that comorbidities may train patients and improve their ability to draw conclusions from healthcare information.

More than half of the participants in this study were identified as having limited health literacy, which according to Gaag et al. [9] means they have difficulties understanding health-related information. In comparison with other gastrointestinal diseases, such as gastrointestinal cancer and inflammatory bowel disease, patients with cirrhosis report lower grades of health literacy [12]. Low understanding of the illness is repeatedly described for patients with cirrhosis [13,15,16,18]. Hence, HCPs need to make an effort to communicate in a manner that enables patients to understand [8], such as using the teach-back technique [10]. A person-centred approach that adjusts educational strategies based on the patient's needs, preferences and abilities could also be beneficial [29]. In a clinical setting this may be implemented by adjusting the transfer of knowledge by means of various available tools such as pictures of medication boxes with labels instead of simply providing oral information about medication and doses. Furthermore, leaflets with large text, as well as the use of illustrations, mind-maps, checklists, reminders and ensuring support from significant others can improve health literacy. In accordance with our previous report [30], this may be achieved by team-based cirrhosis outpatient care with registered nurse continuity, which improved patients' participation in the exchange of information.

## Strengths and limitations

One strength is the size of the study population (n = 167), which makes this study to the largest of its kind [13,14]. Our opposite results to those of Keps et al. [14] and Freundlich Grydgaard and Bager [13] may be explained by differences in patient characteristics. For example, a comparison of Child Pugh scores demonstrate that our cohort had better liver function. The German population [14] had twice as many patients representing Child Pugh group C and the Danish population [13] had a larger group represented by Child Pugh group B compared to the cohort in this study. Another important difference between the present study and the German [14] and Danish [13] ones was the instruments used for measuring health literacy. In contrast to the two previous studies [13,14] that used the health literacy questionnaire that measures patients' lived experiences of nine health literacy related domains [31], we measured functional health literacy using the NVS instrument [22]. While both the NVS and the health literacy questionnaire are validated instruments, they are not fully comparable. The advantage of the NVS is the short time for data collection and providing information on the patient's actual health literacy function, which may be relevant in clinical settings [10,32]. The distribution of the limited *vs* normal health literacy level in our population was close to 50%, which implies a risk of overestimation of the results if reported with the more common odds ratio [33]. The reporting of our results by use of RR may thus reduce the risk of publication

bias. Because the alpha was not adjusted for multiple tests, we recommend that the present results should be interpreted with caution.

During data collection some of the participants perceived the NVS test as an intelligence test, which was experienced as provocative. Therefore, we do not recommend using the NVS for routine screening of health literacy in clinical settings. Instead, HCPs should be observant and ask questions to assess the patients' understanding of the information provided and how they adjust over time. We view the results of this study as a guide for HCPs identifying persons with cirrhosis at risk of not understanding or acting upon health-related information.

## Conclusion

Limited health literacy is common among Swedish adults with cirrhosis. Both covert hepatic encephalopathy and low education might be predictors of limited health literacy. Hence, covert hepatic encephalopathy needs to be actively screened for and optimally treated in outpatient clinics. It is vital that HCPs tailor their patient education based on the patient's literacy level to facilitate understanding, learning and self-management.

## Supporting information

**S1 File. STROBE Statement—checklist: Items that should be included in reports of observational studies.**
(DOC)

**S2 File. Detailed logistic regression analyses of health literacy among patients with cirrhosis.** S2 Table 1 reporting associations between patient characteristics and limited health literacy. S2 Table 2 reporting associations between cirrhosis related disease events and limited health literacy.
(DOCX)

## Acknowledgments

We acknowledge all patients who accepted participation in this study. Further, we are indebted to all RNs who contributed to this study by recruiting patients and collecting data. We extend our thanks to Riccardo LoMartire for statistical consultation. Furthermore, a warm and dedicated appreciation to Fredrik Rorsman, Anncarin Svanberg, Daniel Sjöberg and Elenor Kaminsky for participating in designing the project as a whole.

## Author contributions

**Conceptualization:** Maria Hjorth, Anna Forsberg.

**Data curation:** Maria Hjorth.

**Formal analysis:** Maria Hjorth.

**Funding acquisition:** Maria Hjorth, Anna Forsberg.

**Investigation:** Maria Hjorth.

**Methodology:** Maria Hjorth.

**Project administration:** Maria Hjorth.

**Resources:** Maria Hjorth.

**Software:** Maria Hjorth.

**Supervision:** Anna Forsberg.

**Validation:** Maria Hjorth.

**Visualization:** Maria Hjorth.

**Writing – original draft:** Maria Hjorth.

**Writing – review & editing:** Maria Hjorth, Anna Forsberg.

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
