## [Decision Letter · Decision Letter 0]

23 Jan 2025

Dear Dr. Hjorth,

We look forward to receiving your revised manuscript.

Kind regards,

Isabelle Chemin, PhD

Academic Editor

PLOS ONE

Journal Requirements:

2. In the online submission form, you indicated that [Data are available from Region Dalarna upon reasonable request (e-mail: forsknings.utlamnande@regiondalarna.se) provided that the data can be made available in accordance with applicable data protection and privacy regulations.].

Reviewers' comments:

Reviewer's Responses to Questions

**Comments to the Author**

1. Is the manuscript technically sound, and do the data support the conclusions?

Reviewer #1: Yes

2. Has the statistical analysis been performed appropriately and rigorously?

Reviewer #1: Yes

3. Have the authors made all data underlying the findings in their manuscript fully available?

Reviewer #1: Yes

4. Is the manuscript presented in an intelligible fashion and written in standard English?

Reviewer #1: Yes

Reviewer #1: The authors are commended for conducting the first study of health literacy in a Swedish cirrhosis population. Health literacy is not adequately studied in patients with cirrhosis. In order for the manuscript to be strengthened the following need to be addressed.

1. It is unclear if the patients were counseled by their hepatologists during the recruitment phase of the study spread over 24 months, introducing a bias in the assessment of HL.

2. The inclusion and exclusion criteria need to clearly stated.

3. The methods particularly the Royal Free Hospital – Nutritional Prioritizing Tool (RFH-NPT) section need to be revised for clarity. A correlation between the Royal Free Hospital – Nutritional Prioritizing Tool (RFH-NPT), and HL is not linear, multiple confounders may impact the association, and needs to be justified.

4. Additionally, 70% higher risk of limited health literacy was exhibited in patients with covert hepatic encephalopathy, indicating a significant gap in the management of hepatic encephalopathy despite their symptoms.

5. It will be interesting to know if the same pattern pf limited HL was shown by patients with other GI disease.

**Do you want your identity to be public for this peer review?** For information about this choice, including consent withdrawal, please see our Privacy Policy

Reviewer #1: **Yes: ** Satish Chandrasekhar Nair

---

## [Author Response · Author response to Decision Letter 1]

5 Feb 2025

We thank you and the Editor and Reviewer for constructive comments on our manuscript, which have helped us to improve the text. In addition to the received comments, we have clarified that our analysis concerned previous events of hepatic encephalopathy and not presence of hepatic encephalopathy as we previously stated in the manuscript (page 9, row 162). We have reviewed all comments and suggestions thoroughly. Below we address them point by point. According to the instructions, the revised manuscript is uploaded in two examples 'Revised Manuscript with Track Changes' and ‘Manuscript’, without tracked changes. In line with the made changes, the supporting information have been revised, please see Appendix S1 and S2. We hope you will find our revisions satisfactory.

Kind regards,

The Authors

Editor:

- Thank you for the suggestion for deposit a laboratory protocol. This manuscript thus refers to clinical patient data and do not include any laboratory data. Therefore we find this recommendation not applicable.

Reviewer: Comment 1.

We have added information about each participant regular hepatology consultations, which started at the time of cirrhosis diagnosis (page 5, row 94 to page 6, row 96). This study had a cross sectional design, collecting data at one single time point per patient, we therefore deem the risk of bias due to hepatology consultations being limited.

Reviewer: Comment 2.

Inclusion and exclusion criteria have been clarified with predefined comorbidities that resulted in exclusion. The information have been summarized in a new added table (Table 1), which is placed in the paragraph ‘Participant and setting’. Accordingly the text in the paragraph has been revised. Please see page 5, row 91 to page 6, row 106.

Due to the new added table, the number for the following tables in the manuscript have been revised (Page 10, row 184; page 10, row 190; and page 11, row 198)

Reviewer: Comment 3.

After consideration we have decided to exclude RFH-NPT form the analysis. Consequently, we agree there is a high risk for confounders for this factor. This change has resulted in the following adjustment in the manuscript:

• The paragraph that described the RFH-NPT has been removed from the data collection section (page 8, rows 143-151).

• Risk for malnutrition has been removed from the statistical analysis section (page 9, rows 163-164).

• Risk ratio and confidence intervals for the second generalized logistic regression model have been adjusted in line with the new analysis (page 11, row 196 and Table 4).

• The new analysis revealed a 54 % higher risk for limited health literacy following covert hepatic encephalopathy, instead of previously reported 70 %, (page 11, row 204).

• Reference 28 has been removed, which is visible in the manuscript without track changes.

Reviewer: Comment 4.

We agree that there is a gap in the management of hepatic encephalopathy. Without systematic screening for hepatic encephalopathy, the symptom may not be identified, which is one of our important reported findings in this study. We have revised the text to clarify the problematic situation without screening routines for hepatic encephalopathy (page 12, row 206-207).

Reviewer: Comment 5.

According to Kaps et al 2022 (Reference 12). Health literacy in patients with cirrhosis is reported to be lower than for patients with gastrointestinal cancer or inflammatory bowel disease. In line with this comment, we have elaborated the text for clarification. Please, see page 13, rows 234-236.

---

## [Decision Letter · Decision Letter 1]

11 Mar 2025

Predictive factors for limited health literacy among persons with cirrhosis: A Swedish explorative cross-sectional study

PONE-D-24-54410R1

Dear Dr.  Hjorth,

We’re pleased to inform you that your manuscript has been judged scientifically suitable for publication and will be formally accepted for publication once it meets all outstanding technical requirements.

Kind regards,

Isabelle Chemin, PhD

Academic Editor

PLOS ONE

Additional Editor Comments (optional):

Reviewers' comments:

Reviewer's Responses to Questions

**Comments to the Author**

Reviewer #1: All comments have been addressed

2. Is the manuscript technically sound, and do the data support the conclusions?

Reviewer #1: Yes

3. Has the statistical analysis been performed appropriately and rigorously?

Reviewer #1: Yes

4. Have the authors made all data underlying the findings in their manuscript fully available?

Reviewer #1: Yes

5. Is the manuscript presented in an intelligible fashion and written in standard English?

Reviewer #1: Yes

Reviewer #1: The manuscript is significantly strengthened The queries have been successfully addressed. No further concerns.

**Do you want your identity to be public for this peer review?** For information about this choice, including consent withdrawal, please see our Privacy Policy

Reviewer #1: **Yes: ** Satish Chandrasekhar Nair

---

## [Editor Report · Acceptance letter]

PONE-D-24-54410R1

PLOS ONE

Dear Dr. Hjorth,

I'm pleased to inform you that your manuscript has been deemed suitable for publication in PLOS ONE. Congratulations! Your manuscript is now being handed over to our production team.

Kind regards,

on behalf of

Mrs Isabelle Chemin

Academic Editor

PLOS ONE